# Artificial Neural Network Assisted Multiobjective Optimization of Postharvest Blanching and Drying of Blueberries

**DOI:** 10.3390/foods11213347

**Published:** 2022-10-25

**Authors:** Weipeng Zhang, Ke Wang, Chang Chen

**Affiliations:** 1School of Artificial Intelligence, Beijing Technology and Business University, 11 Fucheng Road, Beijing 100048, China; 2Biological and Agricultural Engineering Department, University of California Davis, One Shields Avenue, Davis, CA 95616, USA; 3Food Science Department, University of Guelph, Guelph, ON N1G 2W1, Canada

**Keywords:** infrared, machine learning, blueberry, multiobjective optimization, genetic algorithm, drying, blanching

## Abstract

This study aimed to optimize the postharvest blanching and drying process of blueberries using high-humidity air impingement (HHAIB) and hot-air-assisted infrared (HAIR) heating. A novel pilot-scale hot-air-assisted carbon-fiber infrared (IR) blanching/drying system was developed. Fresh blueberries with an average diameter of 10~15 mm were first blanched with high-humidity air at 110 °C and 12 m/s velocity for different durations (30, 60, 90, and 120 s); subsequently, the preblanched blueberries were dried at different IR heating temperatures (50, 60, 70, 80, and 90 °C) and air velocities (0.01, 0.5, 1.5, and 2.5 m/s), following a factorial design. The drying time (DT), specific energy consumption (SEC), ascorbic acid content (VC), and rehydration capacity (RC) were determined as response variables. A three-layer feed-forward artificial neural network (ANN) model with a backpropagation algorithm was constructed to simulate the influence of blanching time, IR heating temperature, and air velocity on the four response variables by training on the experimental data. Objective functions for DT, SEC, VC, and RC that were developed by the ANN model were used for the simultaneous minimization of DT and SEC and maximization of VC and RC using a nondominated sorting genetic algorithm (NSGA II) to find the Pareto-optimal solutions. The optimal conditions were found to be 93 s of blanching, 89 °C IR heating, and a 1.2 m/s air velocity, which resulted in a drying time of 366.7 min, an SEC of 1.43 MJ/kg, a VC of 4.19 mg/100g, and an RC of 3.35. The predicted values from the ANN model agreed well with the experimental data under optimized conditions, with a low relative deviation value of 1.43–3.08%. The findings from this study provide guidance to improve the processing efficiency, product quality, and sustainability of blueberry postharvest processes. The ANN-assisted optimization approach developed in this study also sets a foundation for the smart control of processing systems of blueberries and similar commodities.

## 1. Introduction

Blueberry (*Vaccinium corymbosum* L.) is a popular soft fruit grown in broad locations around the world. The presence of bioactive compounds such as ascorbic acid and anthocyanins makes them powerful free radical scavengers [1]. However, fresh blueberries are highly perishable and should be processed to extend their shelf life and off-season availability. As one of the oldest food preservation methods, drying is widely used in preserving seasonal fruit and reducing postharvest losses [2]. The surfaces of blueberries are covered by a waxy hydrophobic layer, which has low moisture permeability and significantly limits mass transfer and slows the drying process [3]. Numerous mechanical, chemical, and thermal pretreatments have been used to overcome this limitation of drying. Chemical dipping and hot water blanching pretreatments can ameliorate the permeability of the berry skin and enhance moisture transfer. However, the chemical residue may lead to food safety risks and cause additional environmental problems with wastewater [4]. In addition, hot-water blanching usually leads to a mass of soluble components lost in the hot water by leaching. High-humidity hot air impingement blanching (HHAIB) is a unique and effective thermal treatment technology that combines the advantages of steam blanching and air impingement technologies. Recently, HHAIB has been used to pretreat peppers [5] and grapes [6]. In a typical HHAIB process, berries are directly impinged by superheated steam at a high velocity for a short duration, which destroys the waxy hydrophobic layer and avoids the loss of water-soluble nutrients [7]. However, the optimal HHAIB conditions for blueberries remain to be determined. 

In order to provide high-quality dried berries to meet the growing demands of health-conscious consumers, multistage or combined drying technologies have been applied for berries. Among them, hot air (HA) drying is one of the most extensively used technologies. However, the low drying efficiency and high energy consumption are not favorable for the quality preservation and sustainability of the process. Hot-air-assisted IR drying offers many advantages, such as low capital investments, greater energy efficiency, and a higher heat transfer rate, which result in reduced drying times and higher drying rates for different products [8]. However, the conventional infrared emitters, such as ceramic emitters, quartz emitters, and gas combustion emitters are usually large in size, complex in configuration, and difficult to install. In recent years, a new type of IR emitter made from carbon-fiber sheets was developed. The thickness of this new IR emitter is only 2 to 4 mm, and it can easily be activated by a regular voltage (220 V/110 V). This new emitter has a relatively simple configuration, a thinner thickness, and a more compact volume, which makes it easy to install in limited spaces such as an HA drying chamber or a vacuum chamber [9]. The proper design of an HAIR drying system that couples the advantages of both HA and carbon-fiber infrared heating may be more efficient compared to the single technologies for the postharvest processing of blueberries.

The main goals of optimizing the postharvest blanching/drying processes for fresh agricultural products are to maximally maintain quality features, including texture and chemical composition, and to minimally consume energy and drying time. Rehydration capability is an important property of dried blueberries, which reflects the irreversible changes in the microstructure and texture properties of food material subjected to different processing conditions. A higher rehydration capability usually indicates lower microstructure damage [10]. Ascorbic acid, a major antioxidant substance in blueberries, is vulnerable to heating. The content of ascorbic acid in dried blueberries is commonly monitored to track the nutrient loss during thermal processes. It is often assumed that if sufficient ascorbic acid is preserved, other nutrients will be preserved as well [11]. Interest in high-quality dried blueberries has increased in recent years, particularly due to its popularity as an ingredient in pizza, breakfast cereals, and various vegetable and fruit dishes [12]. In fact, the optimization of drying conditions is usually multiobjective in nature and involves minimizing the drying time (DT) and specific energy consumption (SEC) and maximizing the ascorbic acid content (VC) and rehydration capacity (RC). Currently, the highly nonlinear relationships between drying parameters and objective functions can be simulated using machine learning approaches, such as artificial neural networks (ANN), which have been proven to be effective in modeling complicated and ill-defined engineering problems [13]. The multiobjective optimization (MOO) problem using an ANN model is to find a vector of decision variables that satisfy the restrictions of all objective functions. MOO problems can be resolved in two different approaches. The typical strategy is to convert other objectives into constraints or to merge two or more objectives into a single objective variable using the weighted sum method. However, the weight selection is easily prejudiced by human subjectivity [14]. 

The other approach for solving MOO problems where multiple objective functions are optimized and obtained under conditions that are in conflict with each other is finding the Pareto-optimal solution. The Pareto-optimal solution is a set of solutions that are not dominated by any other solution in the solution space, where improvement in one objective requires a certain sacrifice of the others [15]. The most critical step in this solving procedure is to find the Pareto front, which contains a series of Pareto optimal solutions within the design space. Once the Pareto front is established, it is simple to choose the best one according to the specific drying requirements. The nondominated sorting genetic algorithm (NSGA II) is a commonly used method that can be used to find the Pareto-optimal solution with a high efficiency based on the ANN model. NSGA-II has been successfully used in the MOO of apple cube drying [16] and dragon fruit slice drying [17]. 

Taking the above into consideration, the objectives of this study were to: (1) develop an HAIR dryer using the new carbon-fiber sheet IR emitters; (2) analyze the effects of the HHAIB blanching time (*BT*), IR heating temperature (*T*), and air velocity (*v*) on the DT and SEC of the drying process and the RC and VC of dried blueberries; (3) develop an ANN model to simulate the nonlinear relationships between the drying variables and the objective functions; and (4) optimize the operating conditions of the blueberry blanching/drying process using the ANN model via the NSGA-II method to simultaneously minimize the DT and SEC and maximize the VC and RC. The outcomes of this study provide an innovative technology for producing high-quality dried blueberries in a more efficient and sustainable way.

## 2. Materials and Methods

### 2.1. Material Preparation

Blueberries (*Vaccinium corymbosum* L. *cv*. *Southern Highbush*) were manually harvested by experienced cultivators in a local organic farm (Fangshan, Beijing, China), immediately transported to the lab, and stored in a refrigerator at 4 ± 1 °C before use. To reduce the effect of the variance in the size and density of the fruits on the blanching/drying characteristics and quality attributes, the harvested blueberries were subjected to a series of size screenings by a vibrating screener (FXJ-LM, Weiming, Shandong, China), followed by density sorting using sodium chloride (NaCl) solutions of different concentrations (30.6 g/L and 62.5 g/L) at 25 °C. Only the mature blueberries with a uniform purplish color, an average diameter of 13 ± 2 mm, and a density over 1041 kg/m^3^ (Figure 1) were selected for the experiments [18]. 

### 2.2. Design of Processing Equipment

Blanching and drying experiments were conducted with laboratory-scale HHAIB equipment (Figure 2) and customized high-precision computer-controlled HAIR equipment (Figure 3), respectively. Both sets of equipment were installed in a Fangshan Yunong Agricultural Technology Co. Ltd. processing workshop (Beijing, China). The details of the HHAIB equipment were reported by Xiao et al. [19] and Zielinska et al. [20]. The HHAIB system contained a steam generator in an air-impingement chamber. The steam was accelerated by a centrifugal fan and impinged through a series of round in-line nozzles. An electrical heating element raised the temperature of the steam to the setting values. Before blanching, the equipment was preheated for 10 min to reach a steady state. Then, a single layer of blueberries was spread on a stainless-steel mesh tray and transferred into the chamber for the treatments. 

A schematic diagram and photos of the HAIR dryer as well as key components are shown in Figure 3. The processing system contained a centrifugal fan (No. 1 in Figure 3) that was installed above the dryer to draw the air into the system and accelerate air into the air distribution chamber. The air velocity could be adjusted by 0~4 m/s by an inverter (EV4300, Taida, Shanghai, China). The oblique air spoiler in the air distribution chamber kept the outlet airflow horizontal, parallel to the sample trays. Infrared heating was provided by carbon-fiber sheet IR emitters (WR6521, Reli, Shanghai, China), as indicated by No. 7 in Figure 3, the detail information of which was thoroughly introduced in a previous study by Zhang et al. [21]. The spacing between two adjacent IR emitters was fixed at 70 mm, and the trays were installed 35 mm below the IR emitters. A thermocouple (SHT25, CLX, Shenzhen, China) with an accuracy of ±0.1 °C was fixed 10 mm blow the carbon-fiber sheet to measure the heating temperature of the IR emitters. The IR heating temperature was controlled by a proportional integral derivative (PID) controller (model E5CN, Omron, Tokyo, Japan). The thermal image embedded in Figure 3 indicates that the heating was uniform over the entire surface of the IR emitter. Before each drying experiment, the HAIR dryer was run for 20 min to achieve a stable temperature.

The energy consumption during the drying processes was measured by a digital power meter (DTSU1717-4P, HangLong, Shanghai, China). Digital load cell systems with a precision of 0.01 g (HYLF-010, Meikong, Hangzhou, China) were installed at the bottom of the system to track the weight change of the samples during the drying process with a 10 min interval. A touch screen (EI18B20, Weinview, Shenzhen, China) was installed on the control system to monitor in real time and record the weight, energy consumption, and other drying parameters.

### 2.3. Sequential HHAIB and HAIR Processing Experiments

To evaluate the effects of blanching time *BT* = {30, 60, 90, 120 s}, IR heating temperature (*T*), and air velocity (*v*) on the quality attributes of blueberries. Natural convection with *v* = 0.01 m·s^−1^, forced convection with *v* = {0.5, 1, 1.5, 2.5 m·s^−1^}, and IR heating temperature with *T* = {50, 60, 70, 80, 90 °C} were selected for experiments following a three-factor full-factorial design with 80 groups of drying experiments in total. According to the preliminary experimental results, the blanching experiments were performed with an air velocity of 12.0 ± 0.5 m/s, a relative humidity of 30 ± 2%, and a blanching temperature of 110 ± 2 °C for different time periods. Subsequently, the HHAIB-treated blueberries were immediately spread in a single layer on a stainless-steel tray with a loading capacity of 4 kg·m^−2^ and transferred to the HAIR dryer. Triplicate experiments were conducted in the same drying conditions. The dried samples were vacuum-sealed in polyethylene bags to prevent moisture absorption and stored in a refrigerator (4 °C) for no longer than 3 days for further analysis.

### 2.4. Drying Time

The sample weight was automatically recorded by the control system. The initial moisture content (*MC**0*) of the dried samples was determined using convective drying at 105 °C for 24 h [22]. The moisture content at drying time *t* (*MC_t_*) was calculated automatically by control system based on Equation (1). The HAIR dryer was stopped automatically when *MC_t_* was less than 0.05 kg/kg in dry basis. Total drying time *t* was recorded by the comprehensive logical judgment of the touch screen terminal.
(1)$$MC_{t}=\frac{M_{t}-M_{0}\cdot (1-MC_{0})}{M_{0}(1-MC_{0})}$$
where *MC_t_* is the moisture content in dry basis at a particular drying time *t*, kg/kg; *M*_0_ is the initial sample weight, kg; and *MC*_0_ is the initial moisture content in dry basis, kg/kg.

### 2.5. Specific Energy Consumption (SEC)

The energy needed to remove 1 kg of water from blueberries was defined as the specific energy consumption (SEC, MJ/kg). The SEC during dehydration was calculated using Equation (2) [23]:(2)$$SEC=\frac{E}{m_{water}}$$
where *E* is the total electrical power consumed in drying, MJ, and *m_water_* is the mass of the moisture removed during drying, kg.

### 2.6. Rehydration Capacity 

For each drying condition, 50 g of dried blueberries were put into a stainless-steel mesh box and immersed in distilled water at 25 °C for 60 min. Tissue papers were used to remove moisture and droplets on the surface before weighing. The following equation was used to estimate the RC [24]:(3)$$RC=\frac{W_{t}}{W_{0}}$$
where *W*_0_ (kg) and *W_t_* (kg) are the weight of the sample before and after rehydration, respectively, kg.

### 2.7. Ascorbic Acid Content 

The VC content in the dried samples under different drying conditions was determined using a method reported by Wang et al. [25] with slight modifications. Specifically, 5.0 g of sample was homogenized with 25 mL of 3% oxalic acid, transferred into a 50 mL volumetric flask, diluted to 50 mL with 3% oxalic acid, and then shaken gently to homogenize the solution. The obtained solution was centrifuged at 4000 rpm for 15 min. Then, 20 mL of supernatant was mixed with 2 g of activated carbon powder, shaken for 1 min, and then filtered. After filtration, 2 mL of clear solution was added to four glass test tubes. Then, 2 mL of a 2% thiourea solution was added to them. One test tube was used as the blank, and 2 mL of 2% 2,4-dinitrophenyl hydrazine(2,4-DNPH) was added to the other three test tubes. All four test tubes were kept at 36 ± 0.5 °C for 3 h in a thermostatic bath and cooled to room temperature. Then, another 2 mL of 2% 2,4-DNPH solution was added with constant stirring, let rest for 15 min, and then transferred to an ice bath for 30 min. After the ice bath, 4 mL of 80% sulfuric acid was gradually added to the test tubes while gentle shaking was applied. The treated test tubes were allowed to rest in the room conditions for 25 min then added to 1 cm cuvettes, and the absorbance of the solution was measured at 520 nm using a spectrophotometer (UV1800PC, HUXI, Shanghai, China). The VC contents in the dried blueberries were calculated with a predetermined calibration curve and reported as mg per 100 g of blueberry sample in dry weight. The analysis was performed in triplicate.

### 2.8. Artificial Neural Network (ANN) Model

A feed-forward backpropagation ANN (BP-ANN) model was constructed to simulate the implicit and nonlinear relationships between the input variables and output variables of blueberry drying. The fully connected ANN model was composed of three layers as shown in Figure 4. (1) the input layer contained three input factors, namely the blanching time (*BT*), IR heating temperature (*T*), and air velocity (*v*); (2) the output layer contained four output factors: the drying time (DT), specific energy consumption (SEC), ascorbic acid content (VC), and rehydration capacity (RC); and (3) one hidden layer with the neuron number to be determined. The input and hidden layer were connected to each other through weights $w_{ij}^{h}$ (*i* = 1, 2, 3; *j* = 1, 2, 3… n). The hidden layer processed the weighted sum of the input variables using nonlinear activation functions. The hidden and output layers were connected to each other through another sets of weights, $w_{jk}^{o}$ (*j* = 1, 2, 3… n; *k* = 1, 2, 3, 4). Various linear or nonlinear activation functions (including Transig sigmoidal, Logsig sigmoidal, and Pureline) were used to perform the transformation between the neurons in the hidden and output layers. Due to the unknown numbers of neurons in the hidden layer and the activation functions, different topologies were examined to discover the most appropriate activation functions and the optimal number of neurons in the hidden layer, denoted by the value of *j*.

The experimental dataset containing 80 groups of conditions (80 × 3 replicates) was randomly divided into three sets, including 70% for training, 15% for testing, and 15% for validation. The classic Levenberg–Marquardt algorithm was used to train the model. The adjusted coefficient of determination ($R_{adj}^{2}$) and the root-mean-square error (*RMSE*) were calculated to evaluate the performance of the ANN model. The qualified fit should have the highest $R_{adj}^{2}$ and lowest *RMSE*. They were given as Equations (4) and (5) [26]:(4)$$RMSE={\left[\frac{\sum {}_{i=1}^{N}{\left(y_{act,i}-y_{pre,i}\right)}^{2}}{N}\right]}^{1/2}$$
(5)$$R_{adj}^{2}=1-\frac{\left(1-\frac{\sum {}_{i=1}^{N}{\left(y_{pre,i}-y_{act,i}\right)}^{2}}{\sum {}_{i=1}^{N}{\left(y_{pre,i}-\bar{y}\right)}^{2}})(N-1\right)}{N-k-1}$$
where $y_{pre,i}$ is the *i*-th predicted output value; $y_{act,i}$ is the *i*-th actual output value; $\bar{y}$ is the mean of the actual output value; *N* is number of observations; and *k* is the number of constants and independent variables in the model, respectively.

### 2.9. Multiobjective Optimization

The purpose of MOO was to determine the optimal HHAIB blanching and HAIR drying conditions (*BT*, *T,* and *v*) that minimize DT and SEC while maximizing VC and RC. The constraints of the problem are also shown in Equation (6). The MOO problem in this study is expressed as follows:(6)$$\mathrm{Objectives}=\{\begin{array}{c} \mathrm{Min}\;\mathrm{DT}(BT,T,v)\\ \mathrm{Min}\;\mathrm{SEC}(BT,T,v)\\ \mathrm{Max}\;\mathrm{VC}(BT,T,v)\\ \begin{array}{l} \mathrm{Max}\;\mathrm{RC}(BT,T,v)\\ 30\leq BT\leq 120\;s\\ 50\leq T\leq 90\;{}^{\circ}C\\ 0.01\leq v\leq 2.5m\cdot s^{-1} \end{array} \end{array}$$

The Pareto front for this MOO problem was generated using a nondominated sorting genetic algorithm (NSGA II) [27]. 

The calculation was implemented in Matlab (Version Mathworks, Math Works Inc., Model-R2018a, Natick, MA, USA). The Pareto-optimal solutions were obtained using the ‘gamultiobj’ function in the MATLAB toolbox. The parameters for the of NSGA II are set and listed in Table 1.

### 2.10. Statistical Analysis

The experimental data are presented as means ± standard deviations (SD). An analysis of variance (ANOVA) was performed to evaluate the influence of different operating conditions on the drying characteristics and quality attributes, followed by a post hoc Duncan’s multiple range test at a significance level of 0.05. A Pearson correlation matrix was used to study the correlations between different processing conditions and response variables. Statistical analyses were performed using SPSS statistics software (version 21.0, SPSS Inc., Chicago, IL, USA).

## 3. Results and Discussion

### 3.1. Drying Characteristics

Figure 5 shows that the DT and SEC were significantly influenced (*p* < 0.05) by the blanching time and drying conditions. The effect of blanching time on the DT of blueberries under a constant drying temperature 80 °C and an air velocity of 1.5 m/s is shown in Figure 5a. The average DTs were 800, 670, 548, and 590 min with blanching times of 30, 60, 90, and 120 s, respectively. The decrease in drying time with the increase in blanching time ranging from 30 to 90 s should be attributed to the disruption of the waxy layer on the surface of blueberries, caused by the high temperature and strong air impingement, which reduced the resistance to moisture transfer during the later drying stages. When the blanching time continued to increase to 120 s, the cell walls in the surface tissue of blueberries were severely damaged, which caused the loss of cell integrity and shrinkage and in return resulted in additional resistance to moisture transfer and drying time. Similar findings were obtained for the HHAIB processing of grapes [28] and apricots [29]. Therefore, a suitable blanching time is important for the pretreatment of blueberries for drying time minimization. Figure 5b shows the DT and SEC under different heating temperatures at the same blanching time of 90 s and air velocity of 1.5 m/s, where increasing the infrared heating temperature reduced the drying time, which should be due to increases in the thermal radiation intensity and the heat and moisture transfer rate, according to the Stephan–Boltzmann law [30]. Figure 5c shows the drying times of blanched blueberries under different air velocities at the blanching time of 90 s and the drying temperature of 80 °C. The average DTs were 654, 602, 548, and 665 min when the air velocities were 0.01, 0.5, 1.5, and 2.5 m/s, respectively. In general, increases in air velocity benefited a reduction in drying time, which should be attributed to the enhancement of the moisture transfer coefficients between the blueberry surface and the air flow [31]. However, it was noticed that further increases in air velocity to 2.5 m/s led to increases in drying times. Such a result should be due to the cooling effect of high-velocity airflow at the blueberry surface. Similar findings were reported by Motevali et al. [32] for the HA drying of mushroom slices and by Nowak et al. [33] for the IR drying of apple slices. 

### 3.2. Specific Energy Consumption

Drying is an extensive energy-consuming process. As shown in Figure 5a, the SEC values were 7.7, 6.0, 4.9, and 6.0 MJ/kg when the blanching times were 30, 60, 90, and 120 s, respectively, when the drying temperature and air velocity were fixed at 80°C and 1.5 m/s. Figure 5b suggests that the SEC decreased with an increase in the drying temperature when the blanching time and air velocity were fixed at 90 s and 1.5 m/s. As for the influence of air velocity, the SEC values were 6.1, 5.3, 4.9, and 7.1 MJ/kg when the air velocities were 0.01, 1,0, 1.5, and 2.5 m/s, respectively, when the blanching time and temperature were fixed at 90 s and 80 °C. The influence of different operating parameters on the SEC followed a similar trend as the drying time. The correlation between the drying time and SEC is shown in Figure 6 with a correlation coefficient of 0.87. According to Equation (2), the SEC was calculated based on the total energy consumption and the mass of moisture removal. As the moisture removal was almost the same for all processing conditions, the SEC should be positively related to the overall energy consumption, which was directly related to the power of the HAIR equipment and the length of the drying time. It was noticed that the correlation was not strictly linear. This was because the power of the heating system was not proportional to the increase in temperature. Among the processing parameters, the drying temperature had the most influence on the SEC, while the air velocity had the lowest impact. Energy consumption is usually one of the most important design and operation parameters in food drying processing. Thus, besides drying time, it is important to consider the SEC in the selection and optimization of the processing conditions. 

### 3.3. Ascorbic Acid Content

The orange columns in Figure 7 illustrate the effects of different processing parameters on the VC content in the dried blueberries. As shown in Figure 7a, the VC contents were 3.60, 4.33, 4.60, and 3.82 mg·100 g^−1^ when the blanching times were 30, 60, 90, and 120 s, respectively, when the heating temperature was fixed at 80 °C and the air velocity was 1.5 m/s. It was noted that when the blanching time increased from 30 to 90 s, the VC content increased and decreased when the blanching time was further elongated to 120 s, showing a reverse trend compared to the drying time. Similar results were reported by Diamante et al. [34] for the hot-air drying of green kiwifruits. From Figure 7b, the VC contents were 1.94, 2.83, 3.71, 4.60, and 3.68 mg·100 g^−1^ when the heating temperatures were 50, 60, 70, 80, and 90 °C, respectively, with a fixed blanching time of 90 s and an air velocity of 1.5 m/s. The highest VC content was obtained with the heating temperature of 80 °C, the blanching time of 90 s, and the air velocity of 1.5 m/s. A further increase in the heating temperature led to a reduction in the VC content. The photos in Figure 7c show the appearance of the dried blueberries at different temperatures. It was found that samples dried at 80 °C maintained the dark blue color of the fresh blueberry, and samples dried at 90 °C had a dark red color. Such phenomena should be attributed to the overheating of the blueberries at 90 °C, which led to the charring of the blueberry surface and a significant deterioration of heat-sensitive bioactive compounds. Similar findings were observed by Nadian et al. [35], who found that the local overheating and scorching of apple slices caused by higher IR temperatures resulted in significantly lower contents of VC. The effect of air velocity on the VC content in the dried blueberries is shown in Figure 7c. Increasing the air velocity had no significant influence on the VC content (*p* > 0.05), which was also verified with the correlation analysis (Figure 6). Similar results were found in the jujube drying process [36].

In fact, the VC contents in dried blueberries were regulated by a series of competing factors. Since ascorbic acid is susceptible to long exposures to heat and oxygen, the VC contents in blueberries were directly related to the drying times and temperatures (Figure 6). On one hand, HHAIB blanching might cause a disruption in the waxy layer on the blueberry surface, facilitated by the moisture transfer and the reduction in drying time, which was beneficial for the preservation of ascorbic acid [37]. On the other hand, the disruption of the waxy layer might have also improved the permeability of oxygen during processing and accelerated oxidation [38]. The results suggest that a suitable blanching time and drying temperature should be selected for the preservation of the bioactive compounds of blueberries.

### 3.4. Rehydration Capacity

The rehydration capacity is a quality index for evaluating the microstructural changes in plant tissues during drying and other treatments [39]. The influence of blanching time on the RC of dried blueberries is shown in the green columns in Figure 7. In general, the influence of blanching time on the RC showed a similar trend as the VC content. The RC values of the samples were 2.34, 2.73, 3.18, and 2.78 when the blanching times were 30, 60, 90, and 120 s, respectively. Samples blanched for 90 s had the highest RC, and further increases in the blanching time led to reductions in the RC. Such results should be attributed to the disruption of the waxy layer on the fruit surface and the microstructural changes in the blueberries. Suitable HHAIB treatments improved the moisture permeability of the fruit surface, which facilitated the moisture absorption during the soaking. However, overblanching led to significant damage to the cell wall, microstructure collapse, severe shrinkage, and case hardening in the blueberries [40], which resulted in lower RC values. Similar to the trend in VC content, the influence of air velocity during the drying process on the RC values of the samples was not statistically significant (*p* > 0.05). Figure 7b shows that the RC increased with an increase in the heating temperature. Such a phenomenon might be because a higher drying temperature led to a higher moisture evaporation rate and thus resulted in higher porosity. Figure 7c shows that increasing the air velocity from 0.01 to 2.5 m/s had no significant effect on the RC at a fixed blanching time of 90 s and a heating temperature of 80. Since the blanching time and drying temperature had significant influences on the RC of the dried blueberries, it was important to optimize the operating parameters of the process for a better RC.

### 3.5. Construction of ANN Model

As shown in the experimental results and the correlation analysis, the three operating parameters were correlated with the four response variables, either positively or negatively, and the coefficients of the correlations were different from each other. In addition, the blueberry blanching and drying processes were complicated in that the influences of different operating parameters on the rates of heat and moisture transfer, quality and microstructure change, and energy consumption could not be easily formulated in an explicit way. Therefore, a feed-forward ANN model with a backpropagation algorithm was suitable to untangle these implicit correlations and was further used for MOO purposes.

To determine the suitable model parameters, including the number of neurons (4, 6, 8, or 10) in the hidden layer, and the activation functions of the hidden layers (‘Transig’ sigmoidal or ‘Logsig’ sigmoidal) and output layers (‘Pureline’ or ‘Logsig’ sigmoidal) of ANN models with different structures were tested. The values of mean square error (MSE) and the coefficients of determination (R^2^) for DT, SEC, RC, and VC under different trained ANN model structures are summarized in Table 2. It was found that the highest R^2^ values for different response variables were achieved under different ANN structures. The highest R^2^ for DT was achieved in group 10 (‘Logsig’ sigmoidal + six hidden neurons + ‘Pureline’); the highest $R_{adj}^{2}$ for RC was achieved in group 1 (‘Transig’ sigmoidal + four hidden neurons + ‘Pureline’); and the highest $R_{adj}^{2}$ for SEC and VC were achieved in group 3 (‘Transig’ sigmoidal + eight hidden neurons + ‘Pureline’). However, it was only in group 3 that the R^2^ values for all four response variables were higher than 0.96 ($R_{adj}^{2}$ = 0.9854, 0.9605, 0.9827, and 0.9889 for DT, SEC, VC, and RC, respectively). The results suggested that a suitable topology is required for the best accuracy of the ANN model. The topology in group 3 was then used for the further analysis. In the current study, an ANN model was developed with a single hidden layer, which is commonly used in food processing studies, and the prediction capability of the model was acceptable. Developing an ANN model with multiple hidden layers might improve the prediction accuracy but with the price of increasing the computation time and is not desired to be integrated into future smart control systems and industrial-scale drying practices.

The prediction capability of the ANN model was evaluated by comparing the experimental data with the prediction values. As depicted in Figure 8a–d, the predicted values for DT, SEC, VC, and RC aligned well with the experimental data. The $R_{adj}^{2}$ and *RMSE* values were 0.996, 0.991, 0.995, and 0.996 and 12.623, 0.447, 0.105, and 0.089 for DT, SEC, VC, and RC, respectively. The results indicated that the ANN model had an acceptable ability to predict the influence of the three operating parameters (*BT*, *T*, and *v*) on the four response variables (DT, SEC, VC, and RC). Thus, it was further used for the MOO of the blanching and drying process of fresh blueberries.

After determining the values of weights ($w_{jk}^{o}$) and biases ($B_{k}$), the response variables were written in analytical equations as Equations (7)–(10):DT = -0.3917*F*_1_+0.3925*F*_2_-0.0601*F*_3_-0.4614*F*_4_-0.5823*F*_5_-0.3100*F*_6_-0.4336*F*_7_+0.0142*F*_8_+0.1802(7)
SEC = -0.6254*F*_1_+0.2887*F*_2_-0.1322*F*_3_-0.3637*F*_4_-0.5103*F*_5_-0.1851*F*_6_-1.2531*F*_7_-0.2409*F*_8_-0.2779(8)
VC = 1.0117*F*_1_+0.1953*F*_2_+0.0236*F*_3_+0.1323*F*_4_+0.7810*F*_5_+0.0119*F*_6_+1.5636*F*_7_+1.2447*F*_8_-0.6516(9)
RC = 0.7483*F*_1_+0.2663*F*_2_+0.0102*F*_3_+0.0050*F*_4_+0.6852*F*_5_-0.0594*F*_6_+0.6438*F*_7_+0.4402*F*_8_-0.5921(10)
where *F_j_* (*j* = 1,2,3,4,5,7,8) were the values of the activation functions of each neuron in the hidden layer, which were obtained as:(11)$$F_{j}=\frac{1}{1+\exp(-2D_{j})}-1$$
where *D_j_* (*j* = 1,2,3,4,5,6,7,8) were the weighted sums of input variables:(12)$${D_{j}}_{\;}=w_{1j}^{h}\times \mathrm{BT}+w_{2j}^{h}\times T+w_{3j}^{h}\times v+B_{j}$$
where *B_j_* was the bias of each neuron in the hidden layer. The determined values of weights $w_{ij}^{h}$ and bias *B_j_* used in Equation (12) for this ANN model are given in Table 3. The weights increased the steepness of the activation function in the hidden layer, which had the power to decide the triggering speed of the activation function, whereas the bias showed the triggering magnitude of the activation function. The objective functions (Equations (7)–(10)) that were determined from the final trained ANN model were used for MOO.

### 3.6. Multiobjective Optimization

An NSGA II method was used to perform the MOO of the blueberry blanching and drying process using the validated ANN model using the set of parameters shown in Table 1. The optimization problem converged to the Pareto optimal set after 126 GA generations. The 30 design points that formed the Pareto front are shown in Table 4. The *BT, T,* and *v* values were in the ranges of 92~96 s, 83.0~89.9 °C, and 0.3~2.4 m/s, respectively. It should be noted that the points on the Pareto front were not superior to each other. Each group point belonging to this front was optimal in the sense that no improvement could be achieved in one component of the objective function that did not lead to a degradation in at least one of the remaining components [40]. The selection of specific optimal sets depends on the main purpose of the process and the sensitivity of equipment control. For example, Group 1 could be selected for the lowest drying time of 366.7 min at a blanching time of 93 s, a heating temperature of 89.1 °C, and an air velocity of 1.2 m/s. Furthermore, a slightly worse result was obtained for Group 4, where the DT and SEC increased by 3.5% and 11.2%, respectively, relative to Group 1, but VC and RC increased by 2.9% and 0.1%, respectively. In addition, the highest VC content of 4.8 mg/100 g (14.6% higher than Group 1) was obtained in Group 13 with a blanching time of 96 s, a heating temperature of 83 °C, and a velocity of 1.15 m/s. Under this processing condition, the drying time was 425.1 min (15.9% longer than Group 1), the SEC was 2.6 MJ/kg (82.8% higher than Group 1), and the RC was 3.25 (3.0% lower than Group 1). Group 20, with a blanching time of 95 s, a heating temperature of 89.9 °C, and an air velocity of 0.34 m/s, led to the lowest SEC of 1.3 MJ/kg (9.1% lower than Group 1), the highest RC of 3.4 (1.5% higher than Group 1), and a similar VC of 4.17 mg/100g (0.5% lower than Group 1), but the drying time was 472.3 min (28.8% longer than Group 1). The results verified the complexity of food processing problems, where multiple response variables could not be reached under the same conditions. The ultimate goal of optimizing the blanching and drying conditions of blueberries is to maximally preserve the quality (that is, to maintain high VC contents and RC in the dried blueberries) with the shortest drying time and lowest energy consumption possible. In Group 13 and Group 20, the highest VC content and RC were achieved but with the cost of significantly longer drying times and larger energy consumptions. In addition, the differences in the optimized conditions of *BT*, *T,* and *v* were minor (less than 2 s, 0.6 °C, and 0.04 m/s, respectively), which were comparable to the sensitivity of the equipment control. Taking these results into consideration, the processing conditions under Group 1 (*BT* = 93 s, *T* = 89 °C, and *v* = 1.2 m/s) were selected as the optimal set.

To validate the optimization results, an additional set of experiments was performed under the optimized conditions. The experimental values of the DT, SEC, VC, and RC were 372 min, 1.46 MJ/kg, 4.08 mg/100 g, and 3.25, respectively (Table 5). The relative error values between the experimental data and the prediction values were 1.43%, 2.06%, 2.70%, and 3.08% for DT, SEC, VC, and RC, respectively, which were considered accurate enough. The findings suggested that the ANN model developed in this study was valid and accurate for the simulation, prediction, and optimization of sequential HHAIB blanching and HAIR drying processes of blueberries for the highest efficiency and preservation of product qualities at a pilot scale. In recent years, with the development of internet of things (IoT) technologies, such as advance sensors and detection techniques, cloud computing, machine learning, and control algorithms, it has become feasible to integrate them into conventional industrial food processing systems. For example, affordable and easy-to-implement sensors (imaging, spectral, acoustic, etc.) that can measure important processing parameters or food quality/safety indicators in real-time can be combined with ANN models to achieve the real-time monitoring and prediction of processing performances. Furthermore, adaptive dynamic programming (ADP) methods can be integrated to achieve the smart control of food drying processes [41]. The methodology developed in this study contributes to the digital transformation of the conventional food industry and can also be expanded to different stages of the food supply chain.

## 4. Conclusions

A novel sequential HHAIB blanching and HAIR drying technology was develop for fresh blueberries, which synergized the advantages of air impingement, steam blanching, and infrared heating. A machine learning model containing a three-layer ANN was established to simulate and predict the influences of the operating parameters (*BT, T,* and *v*) on the DT, SEC, VC, and RC. A multiple-objective optimization regime was developed based on the developed ANN model, a nondominated sorting genetic algorithm (NSGA II), and the Pareto optimization for the simultaneous minimization of DT and SEC and maximization of VC and RC. The optimal operating conditions were determined to be: *BT* = 93s, *T* = 89 °C, and *v* = 1.2 m/s, which led to DT = 372 min, SEC = 1.46 MJ/kg, VC = 4.08 mg/100g, and RC = 3.35. The novel processing technology developed in this study significantly improved the drying efficiency and product qualities, reduced the energy consumption for the efficient and sustainable processing of blueberries, and showed the potential to be transferred to similar commodities. The ANN-assisted prediction and optimization regime sets a foundation for the real-time monitoring and smart control of blueberries and similar food processing systems.

## Figures and Tables

**Figure 1 foods-11-03347-f001:**
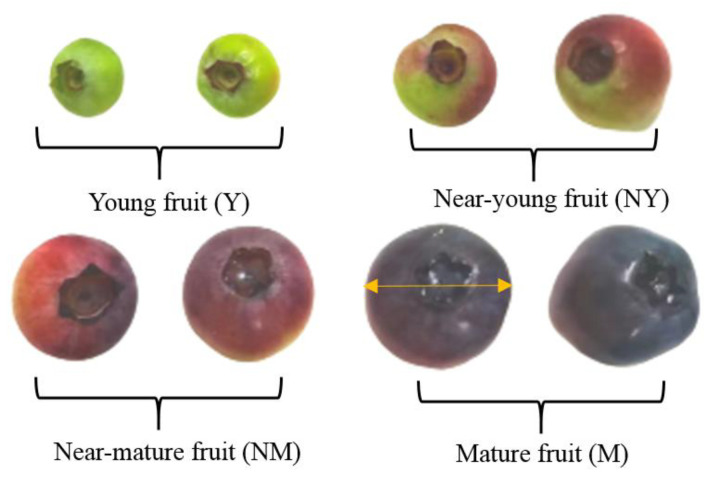
Blueberries that were used in the experimental study.

**Figure 2 foods-11-03347-f002:**
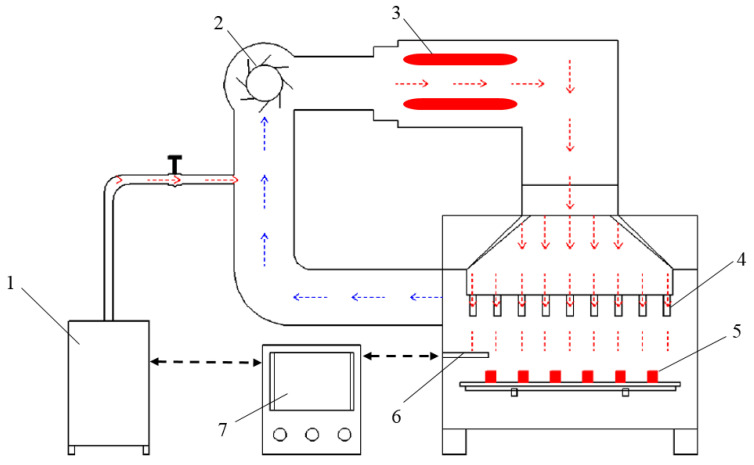
Schematic diagram of HHAIB equipment. (1) Steam generator; (2) centrifugal fan; (3) electric heating assist; (4) impingement round nozzle; (5) material tray; (6) temperature and humidity sensor; (7) control system.

**Figure 3 foods-11-03347-f003:**
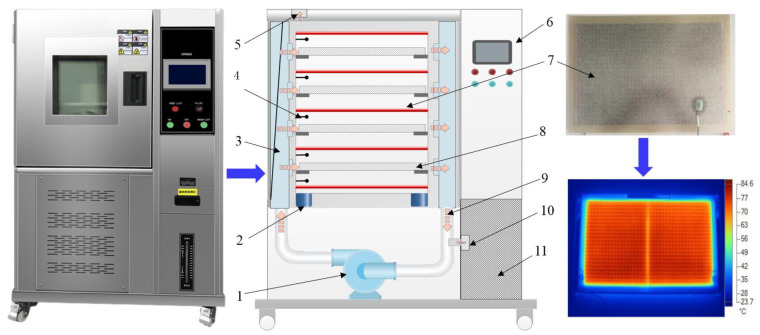
Schematic diagram and photos of HAIR drying equipment. (1) centrifugal fan; (2) load cell; (3) air distribution chamber; (4) temperature sensor; (5) air outlet; (6) control system; (7) infrared carbon-fiber sheet; (8) tray; (9) air flow; (10) air inlet; (11) air filter.

**Figure 4 foods-11-03347-f004:**
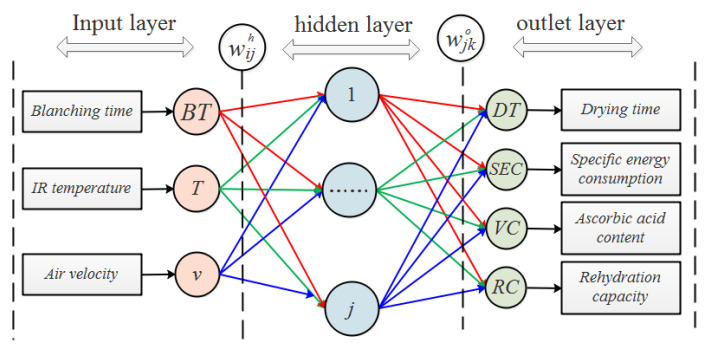
Structure of the ANN model.

**Figure 5 foods-11-03347-f005:**
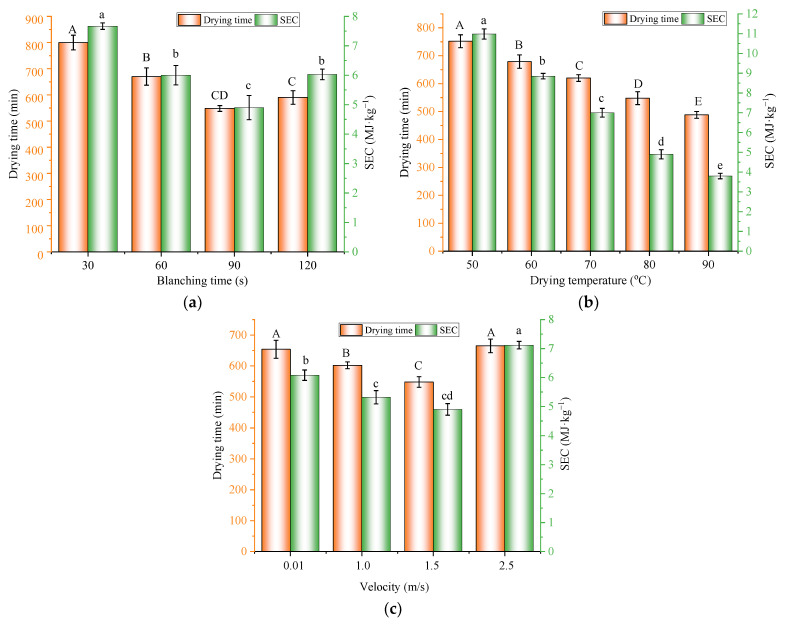
Effect of *BT*, *T*, and *v* on the drying time and SEC for (**a**) an air velocity of 1.5 m/s and a heating temperature of 80 °C, (**b**) a blanching time of 90 s and an air velocity of 1.5 m/s, and (**c**) a blanching time of 90 s and a heating temperature of 80 °C. Note: different characters on the tops of columns of the same color denote significant differences between the mean values (*p* < 0.05).

**Figure 6 foods-11-03347-f006:**
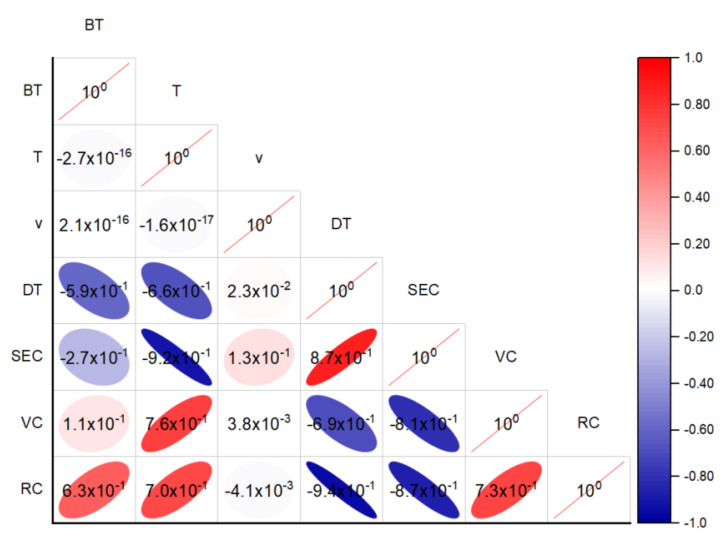
Pearson correlation analysis between different operating parameters and response variables. Note: the numbers in the plots represent the correlation coefficients (r).

**Figure 7 foods-11-03347-f007:**
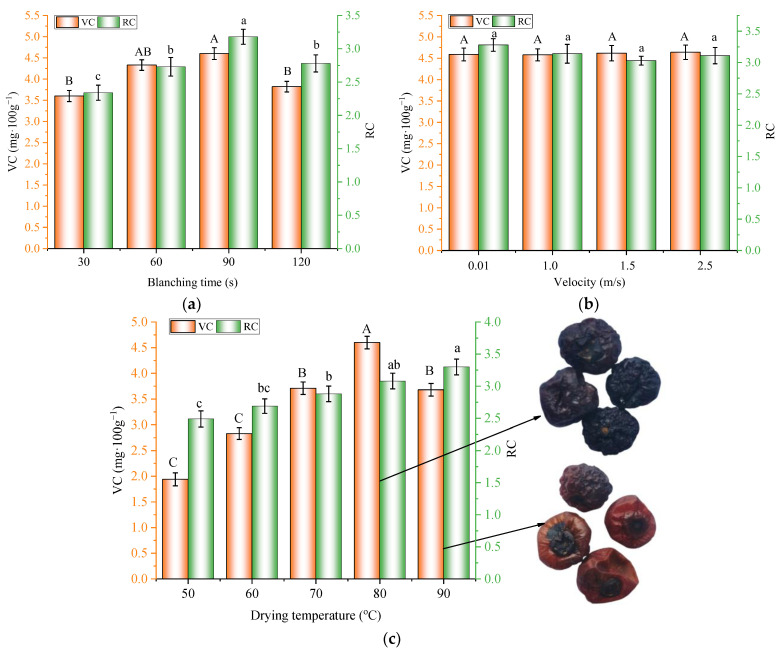
Effects of *BT*, *T*, and *v* on the VC and RC for (**a**) an air velocity of 1.5 m/s and a heating temperature of 80 °C, (**b**) a heating temperature of 80 °C and an air velocity of 1.5 m/s, and (**c**) a blanching time of 90 s and a heating temperature of 80 °C. Note: different characters on the tops of columns of the same color denote significant differences between the mean values (*p* < 0.05).

**Figure 8 foods-11-03347-f008:**
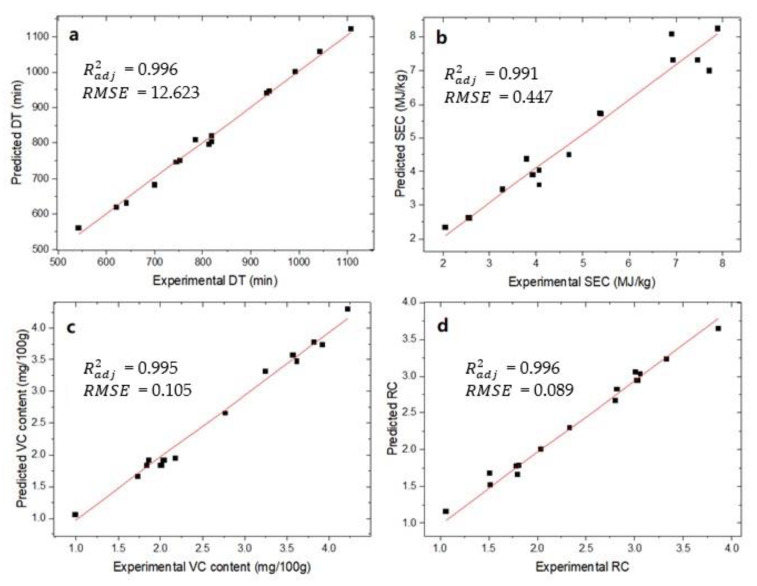
Experimental data and predicted values from ANN model of response variables: (**a**) drying time; (**b**) specific energy consumption; (**c**) ascorbic acid content; and (**d**) rehydration capacity.

**Table 1 foods-11-03347-t001:** Parameters of nondominated sorting genetic algorithm (NSGA II).

Population Type	Double Vector
Population size	30 × number of variables
Crossover function	Intermediate
Crossover rate	90%
Migration function	Uniform
Mutation rate	10%
Number of generations	500 × number of variables
Pareto front population fraction	0.30
Selection function	Tournament size = 2

**Table 2 foods-11-03347-t002:** Test of different ANN structures and activation functions (shaded groups contain the highest R^2^ for at least one response variable; bold font indicates the best solution).

Group	Activation Function of the Hidden Layer	Number of Neurons in the Hidden Layer	Activation Function of the Output Layer	DT		SEC		RC		VC	
				$R_{adj}^{2}\;$	*RMSE*	$R_{adj}^{2}\;$	*RMSE*	$R_{adj}^{2}\;$	*RMSE*	$R_{adj}^{2}$	*RMSE*
1	Tansig	4	Pureline	0.8587	67	0.7969	0.8780	**0.9827**	0.1187	0.8858	0.2042
2		6		0.9935	11	0.8926	0.4324	0.9717	0.1330	0.9263	0.1732
3		8		0.9854	20	**0.9605**	0.4328	0.9773	0.1483	**0.9889**	0.0678
4		10		0.9924	18	0.9438	0.4790	0.9745	0.1261	0.9766	0.1204
5		4	Logsig	0.5721	115	0.4859	1.4736	0.6351	0.5681	0.5237	0.5914
6		6		0.6837	89	0.6270	1.0315	0.7398	0.4637	0.5465	0.5294
7		8		0.9003	50	0.8493	0.8160	0.8121	0.5457	0.6394	0.5531
8		10		0.5853	110	0.6331	1.1254	0.7102	0.4047	0.6745	0.3672
9	Logsig	4	Pureline	0.8066	64	0.8272	0.7444	0.9713	0.1378	0.7638	0.2926
10		6		**0.9952**	12	0.9171	0.5504	0.9734	0.1466	0.8578	0.2423
11		8		0.9871	21	0.9182	0.6151	0.9754	0.1568	0.8901	0.2612
12		10		0.9916	13	0.9143	0.6262	0.9817	0.1217	0.9439	0.1691
13		4	Logsig	0.7066	98	0.6498	1.1809	0.3068	0.6656	0.3069	0.6594
14		6		0.6649	100	0.7267	0.9651	0.6338	0.4124	0.5504	0.2915
15		8		0.5240	107	0.7017	0.9545	0.2826	0.5857	0.4679	0.3550
16		10		0.5659	118	0.6749	1.1610	0.8019	0.3870	0.7407	0.3396
Types of activation functions		$\mathrm{Logsig}=\frac{1}{1+\exp(-n)}$ $\mathrm{Tansig}=\frac{2}{1+\exp(-2n)}-1$ Pureline=n									

Note: The dark background denotes the ANN model structure that was selected for further analysis.

**Table 3 foods-11-03347-t003:** Weights and bias for the CFIR drying of blueberries.

$w_{ij}^{h}$	$w_{1j}^{h}$	$w_{2j}^{h}$	$w_{3j}^{h}$	*B_j_*
1	−0.3826	1.0233	0.1616	0.8245
2	−2.2399	−0.2557	−0.2111	1.3641
3	1.3779	−0.8164	−2.1741	0.1381
4	−0.4903	−0.1129	−1.0111	0.5771
5	1.5830	0.0897	−0.0504	0.8454
6	−0.2677	−0.2317	2.3441	1.2737
7	−0.4421	1.0577	0.1488	−1.5383
8	0.1699	−3.7173	−0.1564	3.7802

*Note: i = 1, 2, 3; j = 1, 2, 3, 4, 5, 6, 7, 8.*

**Table 4 foods-11-03347-t004:** Pareto front optimal solutions for HHAIB and HAIR processing of blueberries.

Pareto ID	BT (s)	T (°C)	*v* (m/s)	DT (min)	SEC (MJ/kg)	VC (mg/100g)	RC
**1**	**93**	**89**	**1.2**	**366.7**	**1.43**	**4.19**	**3.35**
2	93	88.8	1.21	369.7	1.49	4.21	3.35
3	93	88.5	1.24	375.5	1.56	4.24	3.35
**4**	**95**	**88.4**	**1.2**	**379.5**	**1.59**	**4.31**	**3.37**
5	93	86.9	1.16	387.5	1.82	4.4	3.32
6	95	88.7	1.1	389.0	1.53	4.27	3.37
7	95	87.0	1.18	389.9	1.82	4.42	3.34
8	94	85.9	1.14	397.5	2.01	4.49	3.3
9	93	83.2	1.19	407.7	2.5	4.71	3.24
10	94	87.9	0.9	412.8	1.68	4.31	3.34
11	96	83.1	1.21	419.5	2.59	4.79	3.25
12	94	86.6	0.89	423.5	1.92	4.43	3.32
**13**	**96**	**83.0**	**1.15**	**425.1**	**2.62**	**4.8**	**3.25**
14	95	87.6	0.82	430.7	1.76	4.38	3.35
15	95	86.0	1.54	434.5	2.35	4.5	3.32
16	96	83.8	1.51	448.6	2.78	4.72	3.27
17	92	89.8	0.42	457.3	1.31	4.11	3.36
18	93	89.8	0.35	463.0	1.29	4.13	3.37
19	95	85.2	1.73	463.8	2.74	4.58	3.3
**20**	**95**	**89.9**	**0.34**	**472.3**	**1.30**	**4.17**	**3.4**
21	95	84.5	1.85	483.4	3.04	4.65	3.29
22	95	85.5	1.93	486.1	2.91	4.55	3.31
23	95	84.0	1.88	492.2	3.21	4.7	3.27
24	95	84.4	1.99	500.5	3.23	4.66	3.29
25	95	85.1	2.11	510.2	3.22	4.59	3.3
26	95	84.5	2.11	513.7	3.36	4.64	3.29
27	95	84.5	2.16	520.5	3.43	4.65	3.29
28	95	84.8	2.21	524.7	3.44	4.62	3.29
29	95	84.5	2.23	529.5	3.54	4.65	3.29
30	95	84.6	2.35	542.0	3.66	4.64	3.29

Note: The rows with dark background denote the optimum sets in the Pareto front selected.

**Table 5 foods-11-03347-t005:** Prediction and validation results of response variables under the optimal processing conditions.

Results	Operating Conditions			Response Variables			
	*BT* (s)	*T* (°C)	*v* (m/s)	BT (min)	SEC (MJ/kg)	VC (mg/100 g)	RC
Prediction	93	89	1.2	366.7	1.43	4.19	3.35
Validation	93	89	1.2	372	1.46	4.08	3.25
Error (%)				1.43	2.06	2.70	3.08

## Data Availability

All results shown in the manuscript can be requested from the corresponding author.

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
