# Peer review of "Artificial Neural Network Assisted Multiobjective Optimization of Postharvest Blanching and Drying of Blueberries"

_foods, 2022, doi:10.3390/foods11213347_

Round 1
Reviewer 1 Report
The authors here proposed a machine learning model having a 3-layer ANN to describe the relationship between drying conditions (blanching time, drying temperature and gas velocity) and drying time, ascorbic acid content, and rehydration capacity. Overall, the paper is well written and the results of interest for the food industry. Below are some specific comments.
COMMENT #1
A machine learning model containing a 3-layer ANN was estab- 505 lished to simulate and predict the influence of operating parameters (BT, T, v) on the DT, 506 SEC, VC and RC.
COMMENT #2
Throughout the manuscript, I would pay attention to the use of moisture content and water content. The two terms are not synonyms.
COMMENT #3
The authors used an ANN having a single hidden layer. I would add a comment on this choice.
COMMENT #4
LINES 515-516: In the Conclusion section, I would add a comment on how the ANN can be used for process monitoring and control. Which kind of control system can be designed with the support of the ANN outcomes (monitored variables, controlled and manipulated variables, etc..)?
Author Response
The point to point response to the comments are attached.

Reviewer 2 Report
The aim of the study was to estimate the optimal parameters of Blueberries drying conditions using Artificial neural network-assisted multi-objective optimization. Although the manuscript has utilized smart measurement devices and computation techniques, the following concerns need to be addressed.
1. The optimization technique is used in different food preservation even in food drying. The novelty of the manuscript is not strongly established.
2. SEC is not calculated as the waste energy is not considered. Exergy analysis along with SEC to get a real energy utilization perspective should be considered in this optimization study.
3. Overall optimization framework should be presented in an abstract graphic.
4. Drying kinetics needs to be presented with the predicted and validated one.
5. 3D curve representation of multivariable can provide more clear correlations. The author can consider presenting such a multivariate diagram.
6. Uncertainty calculation for drying systems and statistical analysis of modelling needs to be more robust for this type of optimization study.
Author Response
The point to point responses to the comments are attached.

Round 2
